# Design and Analysis of Transformable Wheel with Pivoting-Head Mechanism

**Yaowei Chen \*, Ayumu Kamioka, Masami Iwase, Jun Inoue and Yasuyuki Satoh**

Graduate School of Advanced Science and Technology, Tokyo Denki University, Tokyo 120-8551, Japan
\* Correspondence: chenyaowei@ctrl.fr.dendai.ac.jp

**Abstract:** A transformable wheel designed in this study is proposed to meet the increasingly complex travel requirements of people because it is urgent to provide a smooth and barrier-free travel scheme in complex and changeable urban land. The transformable wheel, with 10 pivoting-head parts, allows vehicles with the wheels to surmount low-height obstacles that deliberately complicates the terrain such as blind roads, small steps, bumps and door sills, and to achieve low-consumption travel. In this study, we demonstrate that the transformable wheels improve its performance by nearly 30% under the road conditions of low-height obstacles and is especially suitable for carts and suitcases passing through low-height obstacles such as blind lanes and low stairs.

**Keywords:** transformable wheel; obstacle surmounting; mechanical analysis; reconfiguration

## 1. Introduction

With social progress and the improvement of living standards, more and more wheeled mechanisms and vehicles are used in a variety of production and life. The fact that their advantages of high speed and high stability greatly improve the operation and transportation efficiency has been well-known. Therefore, most machines, vehicles and mechanisms involving movement equips wheels. However, in the face of irregular roads including bumps, small steps and gravel roads, the driving efficiency of such wheeled machines is reduced. Especially in the case that the wheel radius is relatively small compared to such irregularities, the ability to surmount obstacles and to ride in comfort also deteriorates to different degrees.

At the same time as urbanization [1], in order to meet the various citizens' needs, various terrains, including functional land and connecting channels, are used in urban design. With complex urbanization, we easily find traffic lines such as vehicle roads and sidewalks which are no longer flat, and stairs, steps, slopes, and uneven surfaces in the line. It is easy for a pedestrian to walk on uneven surfaces and steps. However, for users of wheelchairs, silver cars, baby strollers, and walking aids, even small bumps can impede travel movement.

In Asian countries such as Japan, mobility problems are becoming more pronounced due to the aging of the population. With the increasing trend of aging [2–4], a large number of barrier-free facilities are used in urban plan, which greatly improves the travel of the elderly and the disabled. However, there are still many places that are not popularized, and wheelchairs, trolleys and other wheeled mechanisms cannot avoid these places. In addition, in order to cope with some special groups, such as the blind, special infrastructure needs to be built, such as blind roads, which deliberately complicate the terrain and intersperse in the daily functional land.

In view of the above situation, scientists and engineers commonly adopt the method of increasing the wheel radius to improve the ability of surmounting obstacles such as bumps, small steps, door sill etc. The increasing the wheel radius is easily realized, and effect for surmounting obstacles indeed. However, enlargement of wheel radius sometimes

breaks off mechanical specifications such as size, weight and balance. Hence approaches combining the wheel mechanism with other mechanisms have been proposed in [5–8] to achieve the results of reducing the resistance during obstacle surmounting or increasing the obstacle surmounting power.

Zhang et al. [5] presented a wheel-biped transformable robot SR600-II which combines the best of both walking with feet and moving by wheels. They proposed a WtF transformation strategy to tackle the problem when robot transforms from wheeled balance state to in situ biped stance state. It enables the robot to pass by the transition stages in which both wheels and feet touch the ground and to maintain its balance at the same time; R.T. Schroer et al. [6] compared their space robot on wheel-legs, Whegs VP, with cockroach and a wheel robot body motions. It is simple yet effective that Whegs VP can traverse irregular terrain with cockroach locomotion principles. Whegs VP achieves a cockroach-like nominal tripod gait using only a single DC motor, and uses compliant mechanisms in its axles to passively adapt its gait to the terrain such that it can climb obstacles 175% of its leg height; Nagatani et al. [7] developed a "small-sized transformable mobile robot" that is equipped with a mechanism of variable wheel diameter and variable stabilizer length to carry out search and rescue tasks. This robot can be deployed through a narrow space, and it can surmount relatively large obstacles by expanding its wheel diameter to achieve a high capability of moving on bumpy environments and explore narrow spaces; Zarrouk et al. [8] presented a novel highly reconfigurable robot, Rising STAR (RSTAR). RSTAR is fitted with a new four-bar extension mechanism (FBEM) allowing it to extend the distance between its body and legs. This combination of sprawling and extension mechanisms enables RSTAR to overcome extremely challenging obstacles, crawl over flexible and slippery surfaces and even climb vertically in a tube or between two walls.

These aforementioned mobile transport devices and robots can cope with most of the functional land in the city. However, they tend to be designed to adapt to a specific terrain condition, that is, they sometimes encounter a situation they are not suited for, especially for varying complex terrain combined with flat road, bumps, slope, steps etc. Therefore, in order to cope with the above scenarios, transformable wheels have been studied recently which can transform themselves to adapt to the variety of terrain [9–11].

Clark et al. [9] presented a transformable wheel, Adabot, which can smoothly be converted from a round wheel, to a wheel with tire studs, to a legged wheel. Wheel transformations are performed by extending wheel struts radially outward from the center of the wheel. Adabot has been optimized using an evolutionary algorithm such that its physical characteristics and its controller are better able to handle terrain that includes obstacles of varying sizes. Yu et al. [10] presented a novel wheeled robot that transforms from a circled configuration to a spoke-like legged configuration. In general wheeled mobile robots are able to move quickly and efficiently on flat surfaces. The robot's wheel is comprised of five spokes or legs, four of which are actively driven by a motor via four slider-crank linkages. The last one is designed be passive in order to significantly decrease the actuation force of the transformer mechanism. The diameter of the legged configuration wheel is 1.576 times that of the circled configuration; Kim et al. [11] reported a new wheel-leg hybrid robot. This robot utilizes a novel transformable wheel that combines the advantages of both circular and legged wheels. With the new transformable wheel, the robot can climb over an obstacle 3.25 times as tall as its wheel radius, without compromising its driving ability at a speed of 2.4 body lengths/s with a specific resistance of 0.7 on a flat surface.

However, the transformable wheels proposed in these previous works have the following problems: The transformable wheel [9–11] that is based on the round wheel adopts the common method of increasing the wheel diameter to improve the obstacle surmounting performance. At the same time, the shape of the outer side of the round wheel is changed through deformation to show the characteristics of "legs" to further increase obstacle surmounting performance. Especially, as pointed out by [10], there is a good performance in sand, snow field, grassland, leaf land and other dynamic surfaces. However,

for small stepped obstacles such as blind roads, the above research is too overqualified and complicated. The stability of obstacle crossing also decreases greatly with the increasing diameter. It is not suitable for places requiring mobility stability such as welfare care and transportation.

This study proposes a transformable wheel based on the round wheel, which divides the tire equally and changes the rotation angle of the divided tire to reduce the obstacle resistance [12]. When the wheel travels on a plane, a mechanism is used to fix the circumference of the divided tire into a circle, to maintain the stable operation on the plane; when passing through an uneven surface, the divided tire will be rotated to match the irregular terrain surface. Through these mechanisms, we will study a solution that maintains high mobility efficiency on the plane and improves the performance on irregular surfaces.

Benefiting from this structure, the convenience of replacement can be achieved without changing the original wheel size, and it is different from that expanding the radius of wheel to improve the obstacle surmounting performance in the past. Without changing the size, it can focus more on the obstacle surmounting performance of low-height obstacles, and improves its performance ratio, and pay more attention to the performance of load situation.

In the rest of this paper, the mechanism design and working principle of the transformable wheel will be described. The mechanical analysis will be carried out for each case of normal wheel mode and released pivoting-head mode, respectively, with respect to the literature [13,14]. Through experiments, we demonstrate the transformable wheel scheme has good obstacle surmounting performance.

## 2. Structure and Working Principle of Transformable Wheel

A transformable wheel is proposed for obstacle surmounting such as bumps, small steps and door sills found in daily life. The structure of the transformable wheel is illustrated, and its working principle for obstacle surmounting is explained in this section. For the convenience of reading this manuscript, the parameter symbols and their corresponding physical meanings in this paper are laid out in Table 1.

**Table 1.** The parameter symbols used in this manuscript.

| Symbols | Description | Symbols | Description |
|---|---|---|---|
| $r$ | Radius of round wheel | $h$ | Stair height |
| $A$ | Rotation point of segment | $B$ | Contact point |
| $N$ | Pressure on a segment part | $F$ | Friction on a segment part |
| $M_N$ | Pressure torque | $M_F$ | Friction torque |
| $M$ | Wheel torque | $L$ | Distance from the center of the rotation shaft to the stair |
| $a$ | The abscissa of the center of the transformable wheel | $b$ | The ordinate of the center of the transformable wheel |
| $X$ | The abscissa of the center of the contact point | $Y$ | The ordinate of the center of the contact point |
| $\theta_0$ | The angle within friction and stair | $\gamma$ | The rotation angle of segment center |
| $\delta$ | The angle between $OA$ and $X$ axis | $Q$ | Drive force |

### 2.1. Structure

As shown in Figure 1, the transformable wheel in this study adopts a nested structure, and its wheel rim is composed of 10 pivoting-head parts of circular arc. This wheel mainly consists of pivoting-head parts, spoke plates on both sides, a gear-type part, a power spring, steel wire and other components. The pivoting-head parts are sandwiched between the spoke plates and each pivoting-head part can be rotated around an axis which passes through the corresponding tip of the spoke plates. These spokes are coaxial with a gear-type part. The gear-type part controls the rotational degree of freedom of the pivoting-head parts. When the gear-type part fixes the pivoting-head parts, in which case the transformable wheel can be recognized as a normal wheel, the gear-type part and the

spokes rotate simultaneously; with pulling the steel wire, the gear-type part moves axially on the rotating shaft, and releases the fixed pivoting-head parts, in which case the pivoting-head parts rotate along the contacting steps and bumps. At this time, the pivoting-head parts rotates freely with the shape contacting the ground to reduce the resistance during obstacle surmounting; through releasing the steel wire, the gear-type part will be reset by the elastic force of the power spring, and the pivoting-head parts is reset and fixed again to return to the round wheel state.

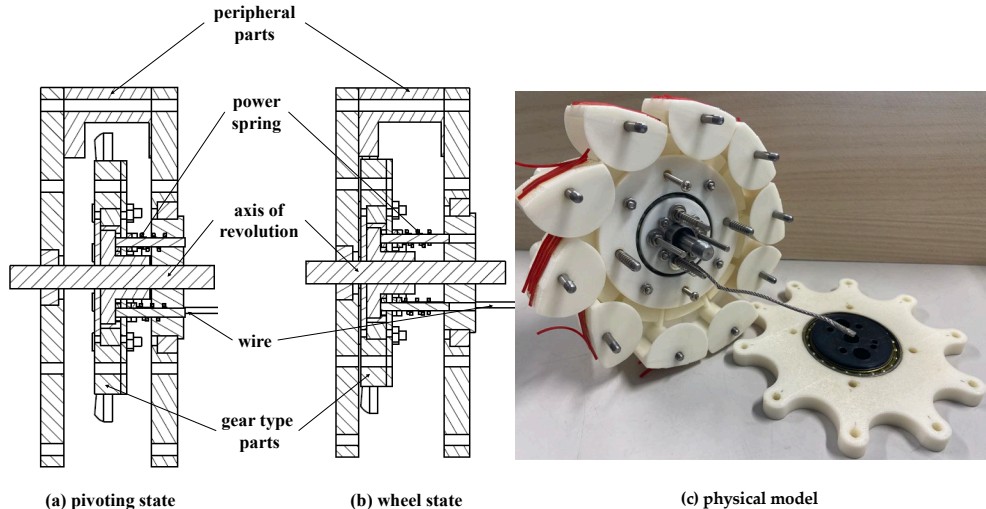

**Figure 1.** Pivot fixing and releasing mechanism ((**a**,**b**) Sectional view; (**c**) Demo).

### 2.2. Working Principle

Before the transformation, the performance of the transformable wheel is equivalent to normal round wheel. In non-planar road conditions such as steps and bumps, the large driving torque is required for the round wheel to obstacle surmounting. If the required driving force of the wheel is greater than the maximum static friction force, the wheel slips.

After the transformation, the pivoting-head parts can rotate freely. According to the contact with the terrain, the pivoting-head parts rotate at a certain angle to fit the terrain. We note that the center of the pivoting-head part deviates from the center of the rotating axis. That is, as different form the circular wheel, the pressure on the pivoting-head part does not pass through the center of the axis, but passes through the center of the pivoting-head part. At this time, the torque generated by the pressure on the pivoting-head part is not zero. Since the wheel torque is composed of the friction torque and the torque of the pressure around the rotating axis, we can reduce the proportion of the friction torque at the condition that the total torque is constant. This principle allows the transformable wheel to reduce the possibility of slipping and improve the obstacle surmounting performance.

## 3. Obstacle Surmounting Performance Analysis

Generally, when surmounting obstacles, the speed of wheel becomes slow, so as to increase the torque to cross obstacles [15]. We used the obstacle height [16,17] to evaluate the mechanism performance of the obstacle surmounting. This study will analyze the obstacle surmounting performance of the state before and after the transformation, with its total torque, pressure torque and friction force of the wheel before and after the deformation.

### 3.1. Analysis of the Round Wheel Obstacle Surmounting Performance

When a round wheel crosses an obstacle, it has a contact point with the obstacle (the object that we discussed in this study is a stair) and the running ground. The center of the rotation axis of the wheel as the origin of a coordinate, the horizontal direction is the *X* axis, and the forward direction of the wheel is positive; and the vertical direction is the *Y* axis, and the gravity direction is positive. The coordinate system is established as shown

in Figure 2. In this coordinate system, the coordinates of the contact point with the stair are $(X_0, Y_0)$. The basic parameter is, stair height, is $h$ the radius of the round wheel is $r$, and the angle between the friction force and the stair plane is $\theta_0$. Based on the above parameters, the following parameters can be obtained:

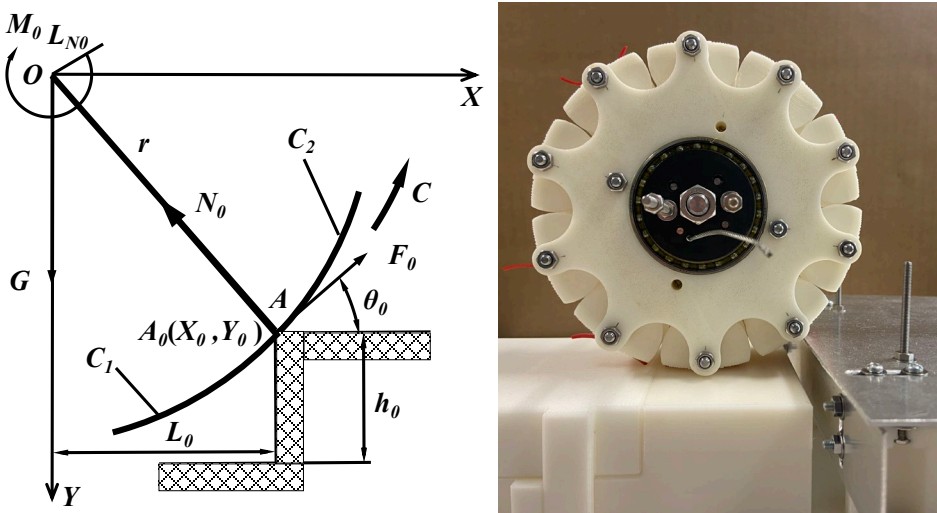

**Figure 2.** Mechanical analysis of round wheel (**Left**: force diagram; **Right**: actual contact).

Pressure on a pivoting-head part $N_0$:

$$N_0 = G \cdot \cos\theta_0 \tag{1}$$

Friction force on a pivoting-head part $F_0$:

$$F_0 = G \cdot \sin\theta_0 \tag{2}$$

Pressure torque $M_{N0}$:

$$M_{N0} = N \cdot L_{N0} = 0 \tag{3}$$

Friction torque $M_{F0}$:

$$M_{F0} = F_0 \cdot r = Gr \cdot \sin\theta_0 \tag{4}$$

Wheel torque $M_0$:

$$M_0 = M_{F0} + M_{N0} = Gr \cdot \sin\theta_0 \tag{5}$$

Distance from the center of the rotation shaft to the stair $L_0$:

$$L_0 = r \cdot \sin\theta_0 \tag{6}$$

Stair height $h$:

$$h = r(1 - \cos\theta_0) \tag{7}$$

When the transformable wheel is in the state of round wheel, it needs enough friction force to make the round wheel complete the obstacle-surmounting action. The maximum friction force provided by the ground contact point is the maximum static friction force $f$. When the maximum static friction force $f$ is insufficient to provide the friction force required for obstacle surmounting, i.e., $F_0 > f$, the wheel will slip and the obstacle surmounting action cannot be completed; on the contrary, when the maximum static friction force $f$ is sufficient to provide the friction force required for obstacle surmounting, that is, $F_0 < f$, the wheel can complete the obstacle surmounting action.

### 3.2. Theoretical Analysis of the Transformable Wheel Obstacle Surmounting

In order to simplify the analysis, only a segment of the pivoting-head parts which contact with the stair is analyzed as a model. As the same way with the analysis of Section 2.1, we establish a coordinate system. The transformable wheel design is the central connection type in this study, and the connect point A divides the segment into two sections $C_1$ and $C_2$. In this study we propose the transformable wheel is a rotary transformation with ten equal segments of the pivoting-head parts. Each equal segment unit as a unified case, and we should discuss the situation of the one equal segment unit. The circumferential angle of a segment unit is 36°, and it will be the same as the previous case every time when it turns 36°. So the symmetrical sections $C_1$ and $C_2$ is 18°. Based on the above situation, the following analysis is available.

As shown in the Figure 3 left, the center line of the segment unit is perpendicular to the ground, which $C_1$ is the left side of center line, and the right side is $C_2$, we call the standard state; as shown in the Figure 3 right, after turning 18° anti-clockwise, the unit side line (dashed line) is vertical to the ground, we call the limit state. The coordinate system is established with the axis of the transformable wheel as the origin of a coordinate, the horizontal direction as the $X$ axis and the vertical direction as the $Y$ axis. Take the $X$ axis as the starting point, and the clockwise direction is the positive. Take the obstacle up to the radius of the transformable wheel as the discussion object, and the segment units contact with ①, ② and ③, with the interval range of $(0, 90°)$. According to the above status, analyzing the height of obstacles can sort out the following Table 2. In Table 2, we mark the standard status as green, and mark the limit status as orange.

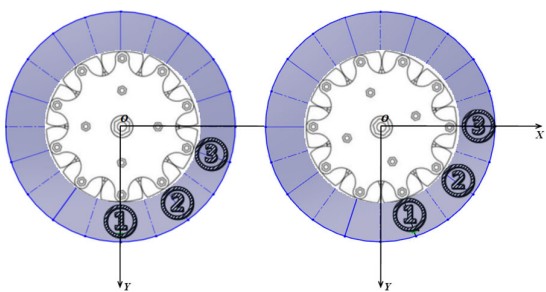

**Figure 3.** Obstacle surmounting posture of transformable wheel (**Left**: standard status; **Right**: limit state; segment unit number: ①, ② and ③, from ground by anti-clockwise).

Under the situation of the actual contact, the parts of the tire that are in contact with the obstacle may be $C_1$ or $C_2$, and in order to facilitate the subsequent analysis and calculation, the contact conditions are only listed in the standard state and the limit state. By integrating the contact conditions in different situations (the contact point is on $C_1$ or $C_2$, and whether the wheel state is the standard or limit state), when dealing with specific obstacle surmounting problems, it can be in the form of a similar way which is look-up table with backward inference method, to infer the situations of wheel contacting, and the limit status (the limit positions of $C_1$ and $C_2$) has an adjustment range of 36°. Based on the posture of the Figure 3, the state reflected in the Table 2 is the opposite. According to the Table 2, for obstacles of the same height, the contact position of the segment is different from the posture of the transformable wheel when contacting. So we can determine the position of the contact segment unit by the height of the obstacle to facilitate subsequent analysis and calculation (the included angle in the next Section is $\delta$). Above it, we can use the parameters, that the specific segment unit contact position and its rotation angle, to improve the range accuracy.

**Table 2.** Correspondence between stair height and segment unit (Orange: the standard state; Green: the limit state; h/mm).

| h/mm | $C_1$ | $C_2$ | h/mm | $C_1$ | $C_2$ | h/mm | $C_1$ | $C_2$ | h/mm | $C_1$ | $C_2$ | h/mm | $C_1$ | $C_2$ |
|---|---|---|---|---|---|---|---|---|---|---|---|---|---|---|
| 1 | | | 14 | | | 27 | | | 40 | | | 53 | | |
| 2 | | | 15 | | | 28 | | | 41 | | | 54 | | |
| 3 | | | 16 | | | 29 | | | 42 | | | 55 | | |
| 4 | | | 17 | | | 30 | | | 43 | | | 56 | | |
| 5 | | | 18 | | | 31 | | | 44 | | | 57 | | |
| 6 | | | 19 | | | 32 | | | 45 | | | 58 | | |
| 7 | | | 20 | | | 33 | | | 46 | | | 59 | | |
| 8 | | | 21 | | | 34 | | | 47 | | | 60 | | |
| 9 | | | 22 | | | 35 | | | 48 | | | 61 | | |
| 10 | | | 23 | | | 36 | | | 49 | | | 62 | | |
| 11 | | | 24 | | | 37 | | | 50 | | | 63 | | |
| 12 | | | 25 | | | 38 | | | 51 | | | 64 | | |
| 13 | | | 26 | | | 39 | | | 52 | | | 65 | | |

### 3.2.1. Mechanical Analysis of Contact Obstacle Surmounting in Section $C_1$

First, we will discuss the situation where the contact point is at $C_1$, as shown in Figure 4. Since the contact point is at $C_1$, the contact point is separated from the connection point $A_1$, then the segment unit deflects, so that the center of the segment is transformed from the origin $O$ to $O_1'$ ($a_1$, $b_1$). At this time, the original connection support $OA_1$ rotates through a certain angle $\gamma$ to become $O_1'B_1$. With the rotation of the segment, the magnitude and direction of the pressure $N_1$ and the friction force $F_1$ will have changed. For ease of comparison, it is assumed that the height of the rotating shaft center $O$ is constant, and the contact point with the stair is $(X_1, Y_1)$. At this time, the angle between $OA_1$ and $X$ axis is $\delta$, and the angle between the friction force and the stair plane is $\theta$. The angle direction starts from the $X$ axis and the clockwise is positive. Based on the above parameters, the following parameters can be obtained:

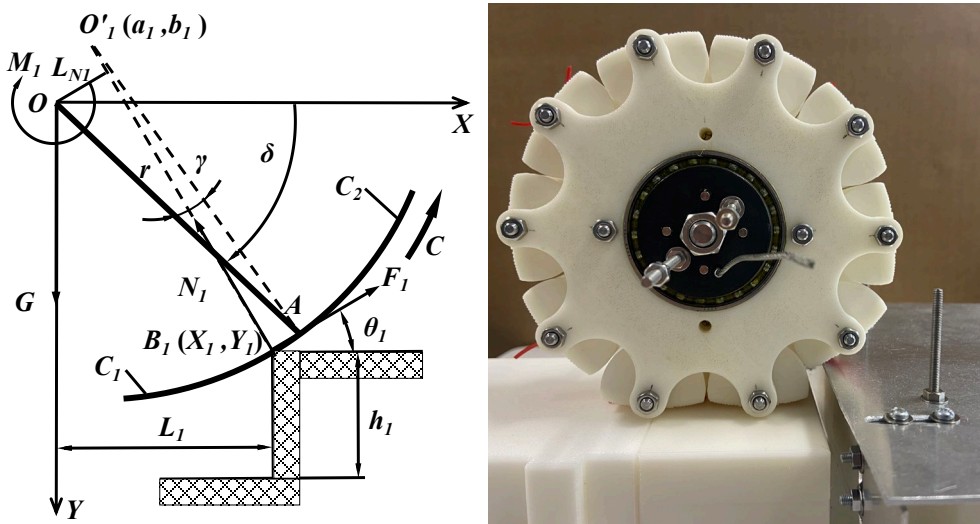

**Figure 4.** Mechanical analysis of $C_1$ section in deformation state (**Left**: force diagram; **Right**: actual contact).

Change distance of center of segment $OO'_1$:

$$\overline{OO'}_1 = 2r \cdot \sin \frac{\gamma}{2} \tag{8}$$

Angle between line $OO'_1$ and $X$ axis:

$$\angle 1 = -\frac{\pi - \gamma}{2} + \delta \tag{9}$$

Coordinate $O'_1$ $(a_1, b_1)$:

$$a_1 = \overline{OO'}_1 \cdot \cos(-\frac{\pi - \gamma}{2} + \delta) = r[\cos \delta - \cos(\gamma + \delta)] \tag{10}$$

$$b_1 = \overline{OO'}_1 \cdot \sin(-\frac{\pi - \gamma}{2} + \delta) = r[\sin \delta - \sin(\gamma + \delta)] \tag{11}$$

Coordinate $B_1$ $(X_1, Y_1)$:

$$X_1 = a + r \sin \theta = r[\cos \delta - \cos(\gamma + \delta) + \sin \theta] \tag{12}$$

$$Y_1 = b + r \cos \theta = r[\sin \delta - \sin(\gamma + \delta) + \cos \theta] \tag{13}$$

From the points $O'_1$ $(a_1, b_1)$ and $B_1$ $(X_1, Y_1)$, the straight line $O'_1 B_1$ equation where the pressure is located can be obtained:

$$Y - b_1 = \cot \theta (X - a_1) \tag{14}$$

The standard form of equation:

$$\sin \theta \cdot y - \cos \theta \cdot x - b_1 \sin \theta + a_1 \cos \theta = 0 \tag{15}$$

According to the distance formula from the point to the straight line, the distance from the origin $O$ to the straight line $O_1'B_1$ can be obtained, $L_{N1}$:

$$L_{N1} = a_1 \cos \theta - b_1 \sin \theta \tag{16}$$

Distance from the center of the rotation shaft to the stair, $L_1$:

$$L_1 = a_1 + r \cdot \sin \theta \tag{17}$$

Stair height $h_1$:

$$h_1 = r(1 - \cos \theta) - b_1 \tag{18}$$

According to the above calculation and force balance, the following torque and friction can be obtained.

Wheel torque $M_1$:

$$M_1 = G \cdot L_1 = Gr[\sin \theta + 2 \sin \frac{\gamma}{2} \sin(\frac{\gamma}{2} + \delta)] \tag{19}$$

Pressure torque $M_{N1}$:

$$M_{N1} = N \cdot L_{N1} = 2Gr \cdot \cos \theta \cdot \sin \frac{\gamma}{2} \sin(\frac{\gamma}{2} + \delta + \theta) \tag{20}$$

Friction torque $F_1$:

$$F_1 = G \cdot \sin \theta \tag{21}$$

### 3.2.2. Mechanical Analysis of Contact Obstacle Surmounting in Section $C_2$

After that, we will discuss the situation where the contact point is at $C_2$, as shown in Figure 5. Since the contact point is at $C_2$, the contact point is separated from the connection point $A_2$, then the segment unit deflects, so that the center of the segment is transformed from the origin $O$ to $O_2'$ $(a_2, b_2)$. At this time, the original connection support $OA_2$ rotates through a certain angle $\gamma$ to become $O_2'B_2$. With the rotation of the segment, the magnitude

and direction of the pressure $N_2$ and the friction force $F_2$ will have changed. For ease of comparison, it is assumed that the height of the rotating shaft center $O$ is constant, and the contact point with the stair is $(X_2, Y_2)$. At this time, the angle between $OA_2$ and $X$ axis is $\delta$, and the angle between the friction force and the stair plane is $\theta$. The angle direction starts from the $X$ axis and the clockwise is positive. Based on the above parameters, the following parameters can be obtained:

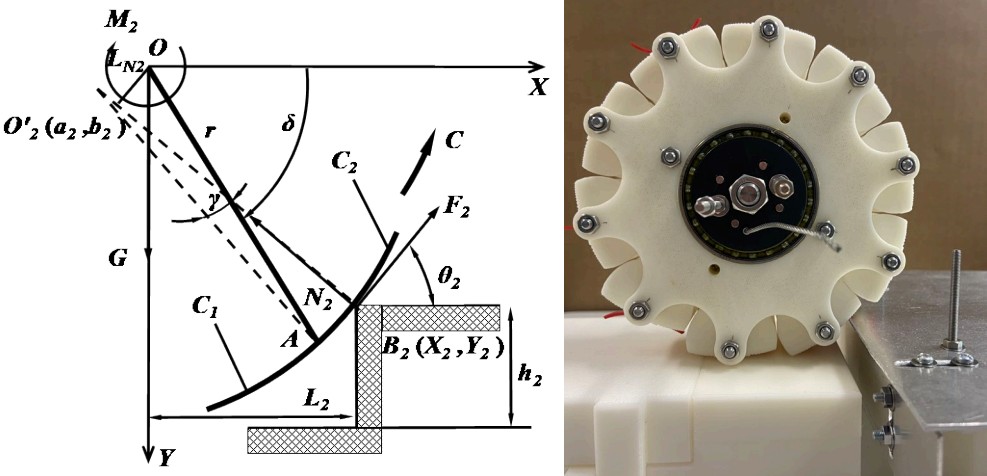

**Figure 5.** Mechanical analysis of $C_2$ section in deformation state (**Left**: force diagram; **Right**: actual contact).

Change distance of center of segment $OO'_2$:

$$\overline{OO'}_2 = 2r \cdot \sin \frac{\gamma}{2} \tag{22}$$

Angle between line $OO'_2$ and $X$ axis:

$$\angle 2 = \frac{\pi - \gamma}{2} + \delta \tag{23}$$

Coordinate $O'_2$ $(a_2, b_2)$:

$$a_2 = \overline{OO'}_2 \cdot \cos(\frac{\pi - \gamma}{2} + \delta) = r[\cos \delta - \cos(\gamma - \delta)] \tag{24}$$

$$b_2 = \overline{OO'}_2 \cdot \sin(\frac{\pi - \gamma}{2} + \delta) = r[\sin \delta + \sin(\gamma - \delta)] \tag{25}$$

Coordinate $B_2$ $(X_2, Y_2)$:

$$X_2 = a + r \sin \theta = r[\cos \delta - \cos(\gamma - \delta) + \sin \theta] \tag{26}$$

$$Y_2 = b + r \cos \theta = r[\sin \delta + \sin(\gamma - \delta) + \cos \theta] \tag{27}$$

From the points $O'_2$ $(a_2, b_2)$ and $B_2$ $(X_2, Y_2)$, the straight line $O'_2B_2$ equation where the pressure is located can be obtained:

$$Y - b_2 = \cot \theta (X - a_2) \tag{28}$$

The standard form of equation:

$$\sin \theta \cdot y - \cos \theta \cdot x - b_2 \sin \theta + a_2 \cos \theta = 0 \tag{29}$$

According to the distance formula from the point to the straight line, the distance from the origin $O$ to the straight line $O_2'B_2$ can be obtained, $L_{N2}$:

$$L_{N2} = a_2 \cos \theta - b_2 \sin \theta \tag{30}$$

Distance from the center of the rotation shaft to the stair, $L_2$:

$$L_2 = a_2 + r \cdot \sin \theta \tag{31}$$

Stair height $h_2$:

$$h_2 = r(1 - \cos \theta) - b_2 \tag{32}$$

According to the above calculation and force balance, the following torque and friction can be obtained.

Wheel torque $M_2$:

$$M_2 = G \cdot L_2 = Gr[\sin \theta - 2 \sin \frac{\gamma}{2} \sin(\delta - \frac{\gamma}{2})] \tag{33}$$

Pressure torque $M_{N2}$:

$$M_{N2} = N \cdot L_{N2} = 2Gr \cdot \cos \theta \cdot \sin \frac{\gamma}{2} \sin(\frac{\gamma}{2} - \delta - \theta) \tag{34}$$

Friction torque $F_2$:

$$F_2 = G \cdot \sin \theta \tag{35}$$

### 3.3. Analysis of the Transformable Wheel Obstacle Surmounting Performance

In order to improve the obstacle surmounting ability of the transformable wheel, the hub connection is connected in the form of a rotating shaft to increase the fitting degree to the obstacles, and to reduce the impact and vibration during obstacle surmounting. The deformation angle is determined by the mechanism design, $\gamma = 27.2°$, as shown in the Figure 6. In addition, the transformable wheel of this study takes low-height obstacles as the discussion object. Among the low-height obstacles, the ability to run over the vertical obstacles can best probe the strength of the ability about the obstacle surmounting performance. Then we select the obstacle height h with 5 mm, 10 mm and 15 mm, and calculate the angle of the wheel state before transforming $\theta_0$, and obtain its value range according to Table 2.

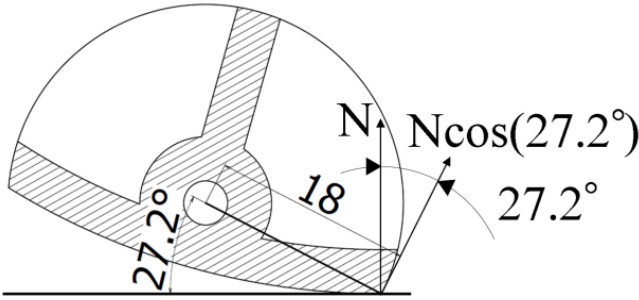

**Figure 6.** Structural design segment angle $\gamma$ ($\gamma = 27.2°$).

### 3.3.1. Value Range

According to the analysis in Section 3.2, beside the constant parameters of structural design such as radius and mass, the final mechanical parameters will be affected by the posture angle $\theta$ and $\delta$. Therefore, according to the existing analysis and constraint conditions, we can obtain value range of the posture angle $\theta$ and $\delta$. During the process of obstacle surmounting, the segment units are not fixed, and it will turn a certain angle with the shape of the obstacle. At this time, the pressure of the transformable wheel from

the obstacle will not pass through the shaft center, and the pressure moment $L_N$ will be generated, as shown in Figures 3 and 4, based on Expressions (16) and (30), we express it as Expression (36). The maximum value of the distance $L$ from the obstacle to the center of the wheel is the radius $r$ of the transformable wheel, based on Expressions (17) and (31), which is expressed as Expression (37). Based on Expressions (7), we need to ensure that there is an actual obstacle colliding with the transformable wheel, and express it as Expression (38). So we can obtain the following constraint condition Expressions (36)–(38):

$$L_N = a\cos\theta - b\sin\theta > 0 \tag{36}$$

$$L = a + r \cdot \sin\theta < r \tag{37}$$

$$\frac{h}{r} = 1 - \cos\theta_0 > 0 \tag{38}$$

Regarding the parameters $a$ and $b$ in the above expression, based on the contact conditions in Figures 3 and 4, select the case where the two inequalities are all true, that is the parameter values at the intersection of the ranges of the inequalities. In Expression (38), since the height of the obstacle in contact will not change suddenly, Expression (7) can be combined with Expressions (18) and (32), then $\theta_0$ can be converted into $\theta$. With the above restraint conditions, Expressions (36)–(38), we can obtain the value range by the coordinate whose the abscissa is $\theta$ and the ordinate is $\delta$, as shown in the red shaded part in Figure 7. Above this, the specific range can be determined according to the obstacle height.

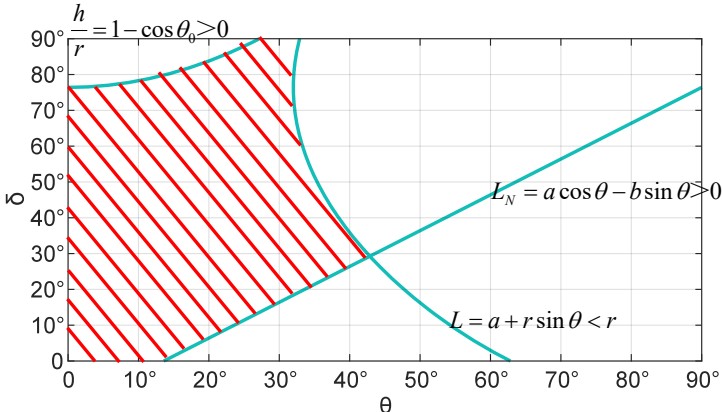

**Figure 7.** Value range of $\theta$ and $\delta$.

### 3.3.2. Theoretical Parameters

According to the analysis in Section 3.2, the wheel torque $M$, pressure torque $M_N$, friction force $F$ and other theoretical parameters can be obtained. The above theoretical parameters are all affected by the posture angle $\theta$ and $\delta$, according to the actual obstacle height, the independent variable can be rewrite into $\delta$, then we can output images with wheel torque $M(\delta)$, Pressure torque $M_N(\delta)$, $F(\delta)$ Etc. The details of the image relationship are as shown in following.

(1)   Torque arm $L$

According to the analysis in Section 3.2, the wheel torque $M$ is affected by the mass of the transformable wheel and the distance $L$ between the axis and the obstacle. The wheel torque $M$ is only related to the contact distance $L$ in different modes. The distance $L$ is smaller, then its torque of obstacle surmounting will be smaller. Based on this, we first compare Equations (6), (17) and (31) when the obstacle height $h$ is 5 mm. Since the radius $r$ is the common factor in the above Equations, so it is compared in the form of $L/r$, and the results are shown in the following.

It can be seen from the Figure 8 that when the obstacle height $h$ is 5 mm, its distance $L_1$ that the contact point in $C_1$ is generally longer than the wheel state distance $L_0$. While

the contact point in $C_2$, is the distance $L_2$ is shorter than $L_0$ as a whole. According to the previous discussion, a case shorter than the wheel state distance $L_0$ can reduce the wheel torque $M$. At this time, the value range of posture angle $\delta$ is $(67.38°, 85.38°)$, so we can only discuss this interval that the contact point in $C_2$.

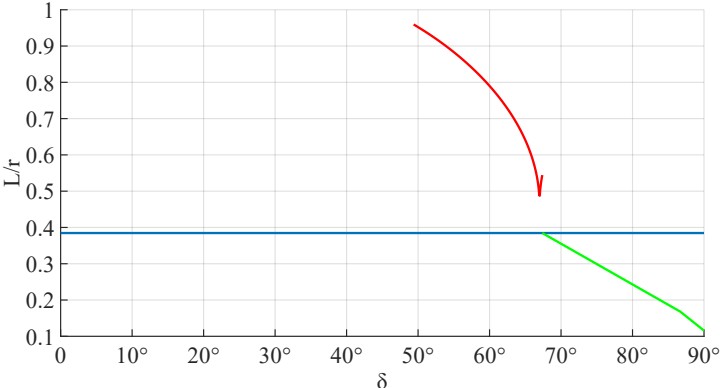

**Figure 8.** Contact distance under transformable state $C_1$ and $C_2$ $L/r$ (5 mm) (Blue: wheel state; Red: $C_1$; Green: $C_2$).

When the obstacle height $h$ is 10 mm as shown in the Figure 9, different from Figure 8, the contact point in $C_2$ is still shorter than the wheel state distance $L_0$ as a whole, while at the contact point in $C_1$, there is a partial part shorter than the wheel state distance $L_0$. At this time, the value range of posture angle satisfying the condition $\delta$ is $[56.72°, 75.8°]$, so we can only discuss this interval. At the same time, the required distance $L_0$ becomes longer as the obstacle height increases.

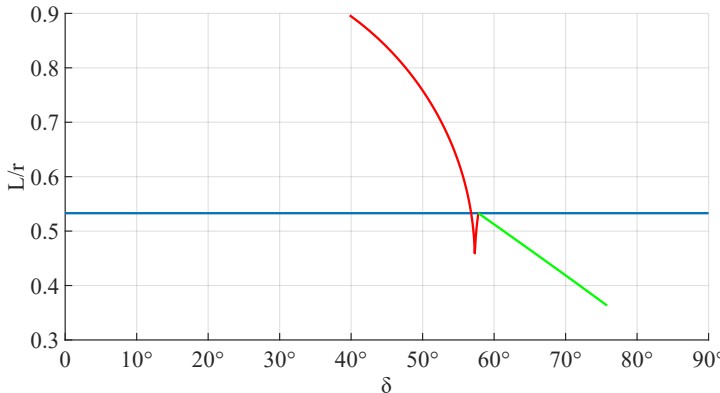

**Figure 9.** Contact distance under transformable state $C_1$ and $C_2$ $L/r$ (10 mm) (Blue: wheel state; Red: $C_1$; Green: $C_2$).

When the obstacle height $h$ is 15 mm as shown in the Figure 10, as same as Figure 9, the contact point in $C_2$ is still shorter than the wheel state distance $L_0$ as a whole, while at the contact point in $C_1$, there is a partial part shorter than the wheel state distance $L_0$. At this time, the value range of posture angle satisfying the condition $\delta$ is $(42.65°, 68.28°)$, so we can only discuss this interval. At the same time, the required distance $L_0$ becomes longer as the obstacle height increases.

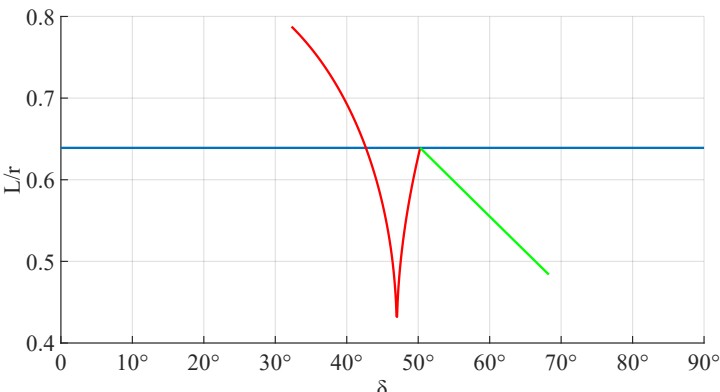

**Figure 10.** Contact distance under transformable state $C_1$ and $C_2$ $L/r$ (15 mm) (Blue: wheel state; Red: $C_1$; Green: $C_2$).

In conclusion, above three situations, the contact distance $L_2$ is shorter than its wheel state $L_0$. In the case of extremely low-height obstacle, $L_1$ is longer than $L_0$. With the increase of obstacle height, there is a partial part $L_1$ will be shorter than $L_0$.

(2)    Wheel torque $M$

Through the analysis of the torque arm $L$, we can only analyze the contact point in $C_2$ and a part of $C_1$. By eliminating the posture angle $\theta$ to obtain the wheel torque function $M(\delta)$. At the same time, in order to facilitate the comparison with the wheel state, the relationship is further rewritten as function $M/M_0(\delta)$. The relationship is as follows Equations (39) and (40). Then we substitute the obstacle heights to obtain the respective visual images.

$$\frac{M_1}{M_0} = \frac{\sqrt{1 - [\cos\theta_0 + \sin(\gamma + \delta) - \sin\delta]^2} + \cos\delta - \cos(\gamma + \delta)}{\sin\theta_0} \tag{39}$$

$$\frac{M_2}{M_0} = \frac{\sqrt{1 - [\cos\theta_0 - \sin(\gamma - \delta) - \sin\delta]^2} + \cos\delta - \cos(\gamma - \delta)}{\sin\theta_0} \tag{40}$$

It can be seen from Figure 11 that when the obstacle height $h$ is 5 mm, the projected situation is as same as that in Figure 8, and the situation of contact point in $C_2$ can reduce the torque required for obstacle surmounting. In the minimum case, it can be reduced to less than 50%.

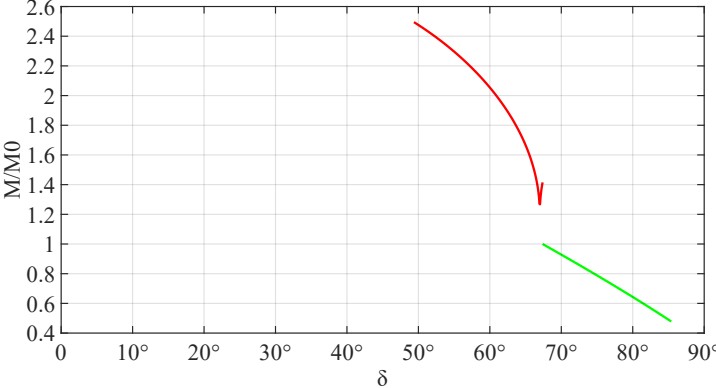

**Figure 11.** The obstacle height h is 5 mm. Comparison of torque under transformable state $C_1$ and $C_2$ $M/M_0$ (Red: $C_1$; Green: $C_2$).

When the obstacle height $h$ is 10 mm as shown in the Figure 12, different from Figure 11, the projected situation is as same as that in Figure 8, and a part of curve the contact point

in $C_1$ is less than 1. However, the overall torque of $C_1$ is greater than that of $C_2$. Under the limit condition of $C_1$, the torque can be increased by 70%, and the minimum torque can be reduced by 15%; while it is in $C_2$ it can be reduced by about 30%.

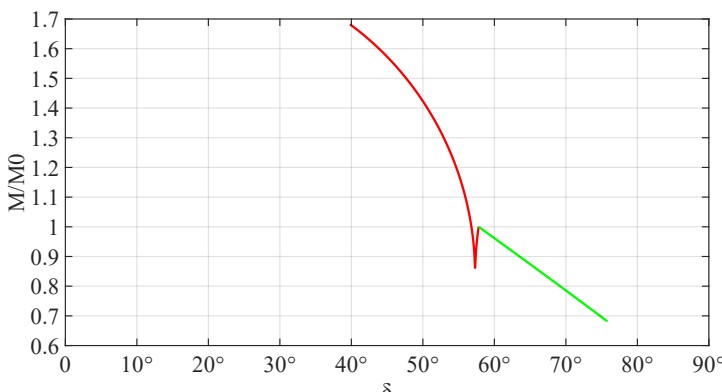

**Figure 12.** The obstacle height h is 10 mm. Comparison of torque under transformable state $C_1$ and $C_2$ $M/M_0$ (Red: $C_1$; Green: $C_2$).

When the obstacle height $h$ is 15 mm as shown in the Figure 13, as same as Figure 12, there is still a part of the curve for which the the contact point in $C_1$ is less than 1, and the span of $C_1$ has been larger than that of $C_2$. Under the limit condition of $C_1$, the torque can be increased by 22%, and the minimum torque can be reduced by 32%; while it is in $C_2$ it can be reduced by about 15%.

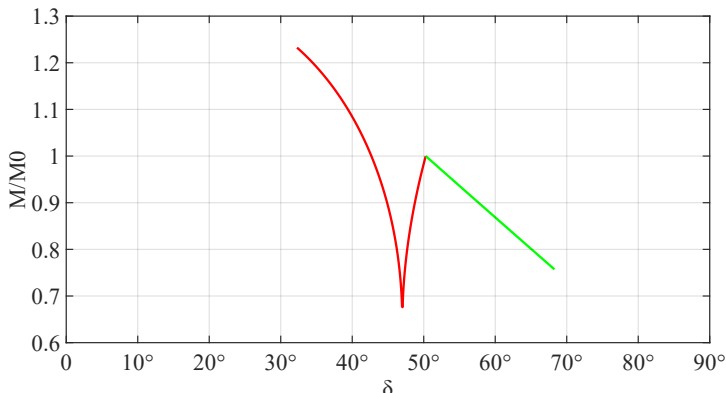

**Figure 13.** The obstacle height h is 15 mm. Comparison of torque under transformable state $C_1$ and $C_2$ $M/M_0$ (Red: $C_1$; Green: $C_2$).

Through the comparison with the torque required for the transformable wheel to surmount the obstacle in the wheel state, it can be clearly concluded that the torque decreases in the transformable state, and it has a quantitative analysis. The wheel torque in the $C_2$ is always reduced, and it can run over without increasing power. However, with the increase of the obstacle height, the reduction range becomes lower and lower; In addition to the extremely low-height, the torque in $C_1$ will also be reduced. With the increase of obstacle height, the increase of wheel torque will decrease and its reduction will increase. With the obstacle height increasing, the change will be more obvious. Based on the above analysis, the transformable wheel has better obstacle-surmounting performance in terms of power.

(3)    Drive force $Q$

According to the previous analysis, the requirements for obstacle surmounting, such as the wheel torque $M$ and the corresponding torque arm $L$, can be obtained. In order to provide the required wheel torque $M$, it is necessary to provide the drive force $Q$ to drive the

transformable wheel to surmount the obstacle. According to the analysis of Figures 4 and 5, the function $Q(\delta)$ of corresponding contact condition can be obtained.

$$Q_1 = G \cdot \frac{\sqrt{1 - [\cos \theta_0 + \sin(\gamma + \delta) - \sin \delta]^2} + \cos \delta - \cos(\gamma + \delta)}{\cos \theta_0} \tag{41}$$

$$Q_2 = G \cdot \frac{\sqrt{1 - [\cos \theta_0 - \sin(\gamma - \delta) - \sin \delta]^2} + \cos \delta - \cos(\gamma - \delta)}{2 \sin \delta + 2 \sin(\gamma - \delta) + \cos \theta_0} \tag{42}$$

The image information in Figures 14–16, which is consistent with the described situation in the section of torque arm $L$ and wheel torque $M$. By observing the figures in Section 3.3.2, the red curves which is contacted with $C_1$ all appear V-shape, and combined with the corresponding calculation formula, it is found that the contact section of $C_1$ has the composition of Equation (43). After calculation, there is a situation that the formula under root will be less than 0, so we simplify it to the size of 1 and $\sin\theta_0 + \sin(\gamma + \delta) + \sin\delta$. In its equivalent case, it is the extreme value of the V-shaped inflection point. At the same time combining Figure 4 and Equation (11), Figure 17 can be obtained. The actual meaning of Equation (43) is the comparison of $s_1$ and $s_3$. When the two are the same, it is the extreme value of the inflection point. At the same time, when the height $h(s_1)$ is low, the proportion of $s_2$ is relatively large, and the segment only needs to rotate slightly to generate the $s_3$, that is, as shown in Figures 8, 11 and 14, the distance between the extreme points and the connection points by red and green curves is close; on the contrary, when the height $h(s_1)$ becomes larger, the proportion of $s_2$ becomes smaller, and the rotation of the segment will become larger, so that the corresponding $s_3$ can be generated, that is, as shown in Figures 10, 13 and 16, the distance between the extreme points and the connection points by red and green curves is far.

$$\sin \theta = \sqrt{1 - \cos^2 \theta} = \sqrt{1 - [\cos \theta_0 + \sin(\gamma + \delta) - \sin \delta]^2} \tag{43}$$

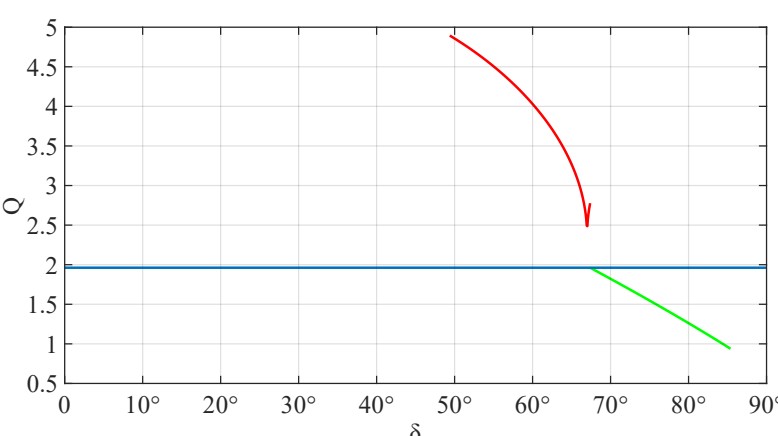

**Figure 14.** Drive force under transformable state $C_1$ and $C_2$ $Q$ (5 mm) (Blue: wheel state; Red: $C_1$; Green: $C_2$).

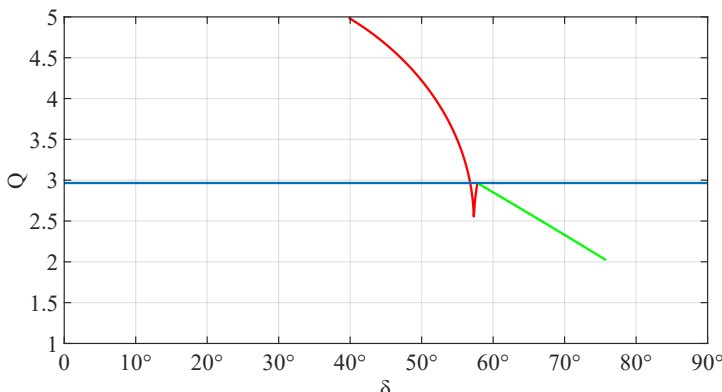

**Figure 15.** Drive force under transformable state $C_1$ and $C_2$ $Q$ (10 mm) (Blue: wheel state; Red: $C_1$; Green: $C_2$).

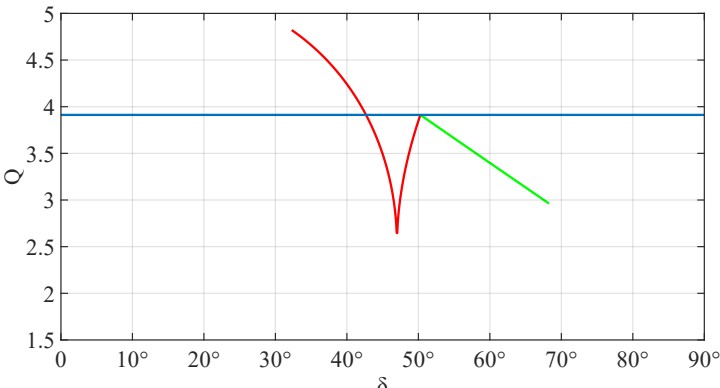

**Figure 16.** Drive force under transformable state $C_1$ and $C_2$ $Q$ (15 mm) (Blue: wheel state; Red: $C_1$; Green: $C_2$).

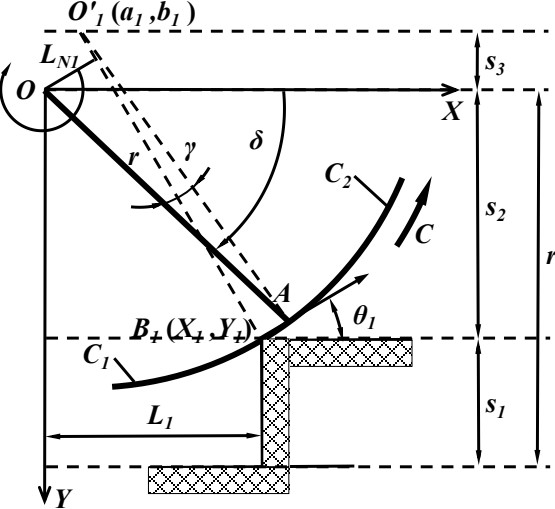

**Figure 17.** The physical meaning of extreme points_(4).

On this basis, we carry out a joint discussion with the experiment of the previous study [7,12,18].

In this experiment, the transformable wheel is placed on a bench with variable height, and its rotating shaft is connected with a tension gauge, as shown in Figure 18. In the case of sufficient friction, the horizontal pull can be regarded as its Drive force. When we pull the force gauge horizontally, the transformable wheel will surmount the obstacle, and we

record the data on the force gauge at this time. The force gauge used in this experiment is AIKOH Model 9800 Series as shown in Figure 19. Its measurement accuracy is ±0.2%F.S, and the smallest unit of measurement is 0.01 N, and the measurement range is 0~50 N. We record the required the force of the wheel state and the transformable state in Table 3.

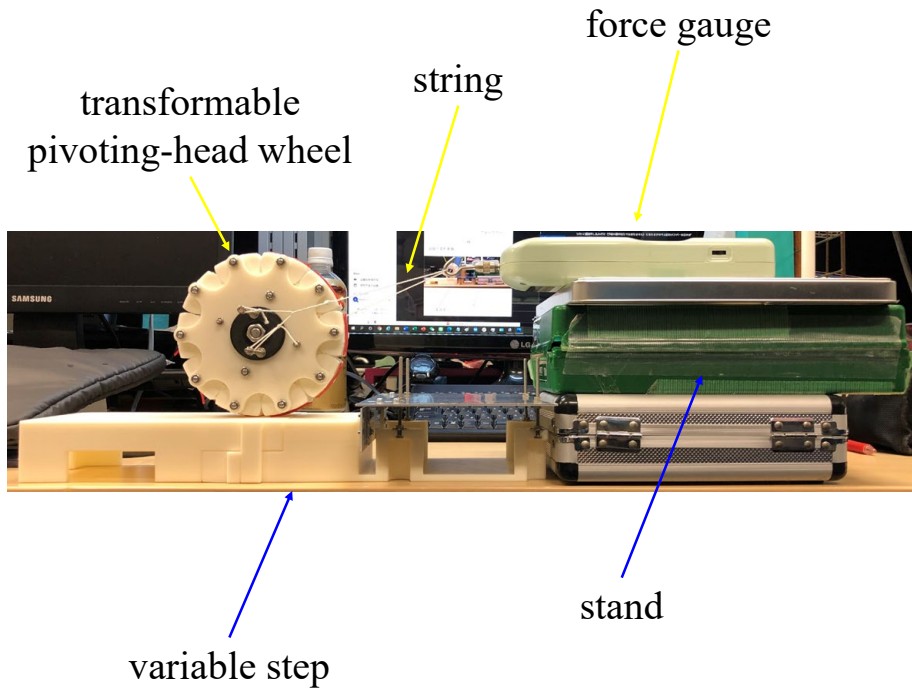

**Figure 18.** Experimental environment for measuring the force to overcome a step.

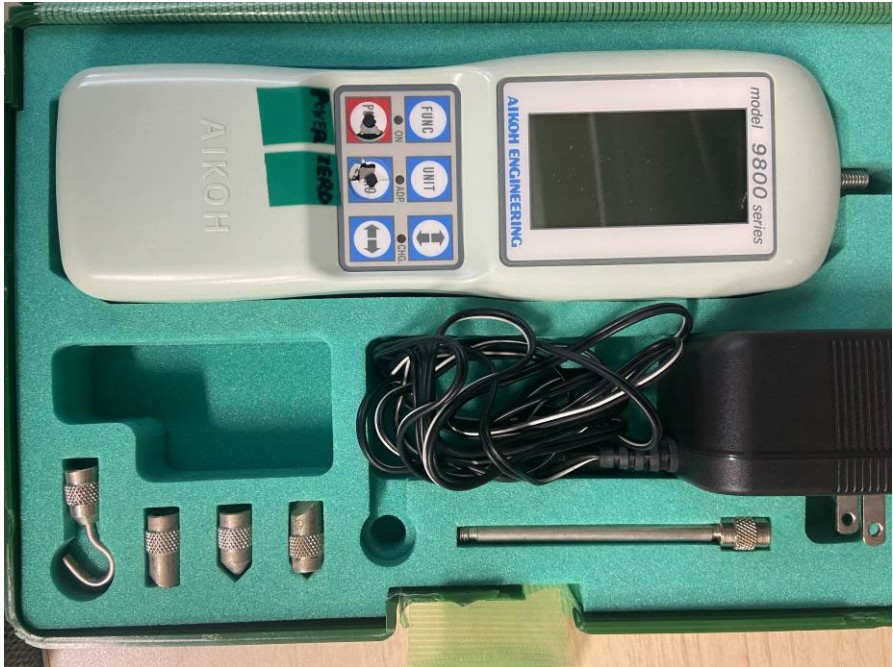

**Figure 19.** Force gauge AIKOH Model 9800 Series.

**Table 3.** Step overcoming force of the wheel state according to the height of the step.

| | | Step Overcoming Force in Wheeled State [N] | | | | | | Step Overcoming Force in Transformable State [N] | | | | | |
|---|---|---|---|---|---|---|---|---|---|---|---|---|---|
| | | 1st | 2nd | 3rd | 4th | 5th | Avg. | 1st | 2nd | 3rd | 4th | 5th | Avg. |
| | 0 | 0.72 | 0.58 | 0.58 | 0.50 | 0.52 | 0.58 | 1.68 | 1.71 | 1.65 | 1.71 | 1.73 | 1.70 |
| | 1 | 0.88 | 0.81 | 0.58 | 0.85 | 0.69 | 0.76 | 1.78 | 1.78 | 1.74 | 1.89 | 1.78 | 1.79 |
| | 2 | 1.01 | 0.98 | 1.17 | 1.20 | 1.32 | 1.14 | 1.90 | 1.94 | 1.94 | 1.84 | 1.87 | 1.90 |
| | 3 | 1.48 | 1.47 | 1.52 | 1.49 | 1.65 | 1.52 | 1.78 | 1.95 | 1.87 | 1.86 | 2.12 | 1.92 |
| | 4 | 1.68 | 1.75 | 1.61 | 1.81 | 1.80 | 1.73 | 2.06 | 2.18 | 2.05 | 1.91 | 2.20 | 2.08 |
| | 5 | 2.00 | 1.94 | 2.12 | 2.06 | 2.03 | 2.03 | 2.34 | 2.15 | 2.28 | 2.08 | 1.93 | 2.16 |
| Step height [mm] | 6 | 2.21 | 2.18 | 2.08 | 2.16 | 2.19 | 2.16 | 2.21 | 2.15 | 2.45 | 2.43 | 2.11 | 2.27 |
| | 7 | 2.28 | 2.38 | 2.40 | 2.14 | 2.27 | 2.29 | 2.17 | 2.57 | 2.25 | 2.59 | 2.30 | 2.38 |
| | 8 | 2.56 | 2.51 | 2.48 | 2.43 | 2.31 | 2.46 | 2.40 | 2.44 | 2.58 | 2.64 | 2.70 | 2.55 |
| | 9 | 2.79 | 2.81 | 2.72 | 2.83 | 2.77 | 2.78 | 2.86 | 2.80 | 2.77 | 2.60 | 2.48 | 2.70 |
| | 10 | 2.84 | 2.99 | 2.96 | 2.94 | 2.90 | 2.93 | 2.81 | 2.91 | 2.95 | 2.77 | 2.61 | 2.81 |
| | 11 | 3.09 | 3.02 | 3.06 | 2.98 | 3.14 | 3.06 | 3.09 | 2.70 | 3.22 | 3.19 | 2.65 | 2.97 |
| | 12 | 3.51 | 3.21 | 3.16 | 3.26 | 3.15 | 3.26 | 3.30 | 3.08 | 2.53 | 3.26 | 2.67 | 2.97 |
| | 13 | 3.51 | 3.30 | 3.17 | 3.42 | 3.36 | 3.35 | 3.38 | 3.26 | 2.76 | 3.32 | 3.14 | 3.17 |
| | 14 | 3.46 | 3.42 | 3.61 | 3.59 | 3.45 | 3.51 | 3.34 | 3.35 | 2.81 | 3.33 | 2.91 | 3.15 |
| | 15 | 3.63 | 3.67 | 3.58 | 3.61 | 3.53 | 3.60 | 3.35 | 3.40 | 2.90 | 3.47 | 2.98 | 3.22 |

We compared the blue line of the wheel state in Figures 14–16 with the average tension in Table 3 with the corresponding height, and found that the data at 5 mm and 10 mm are basically consistent; there is a big difference from the data of 15 mm, with an increase of about 8.3%. It is speculated that the wheel surface friction is insufficient, resulting in the wheel slipping. On the basis of the wheel state in Table 3 will be compared.

Based on Table 3, the 5 mm average value is 2.16 N, but there is no corresponding value in Figure 14. Here, we discuss the reason why the curves of $C_1$ and $C_2$ are not connected under the same function, different from Figures 15 and 16. In Equations (41) and (42), there is a part of root operation, which is the part of posture angle $\sin\theta$. In Figures 14–16, the red curve of $C_1$ has a "V" shape. After calculation, at its turning point, the $\sin\theta$ is 0 ($\sin\theta = 0$), then $L = a$, $L_N = a$, i.e., $O'A$ is perpendicular to the ground. In this posture, a minimum drive force is required. In Figures 14–16, the minimum force is about 2.5 N, and its value slightly increases with the elevation of the obstacle. Interestingly, in Figure 20, the power demand in the transformable state starts to decrease just after the 8 mm step with the power of 2.46 N in the wheel state. It can be seen that the current analysis cannot explain the situation of steps below 8 mm under the specifications of transformable wheel in this study.

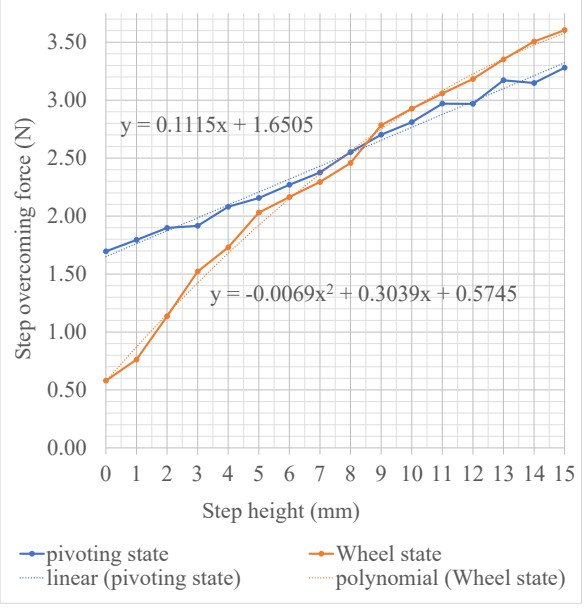

**Figure 20.** Graph of strength to get over a step.

Above 8 mm steps in Figures 15 and 16, we can find the corresponding point to 2.81 N in 10 mm and 3.6 N in 15 mm. At this time, there are two posture angles in $C_1$ and one posture angle in $C_2$, which meet the above conditions. Through the actual contact situation, the corresponding posture angle can be obtained.

## 4. Conclusions

The transformable wheel in this study is developed by the characteristic that the pivoting-head parts generate the torque such that the pressure does not pass through the wheel axis by the wheel transforming. Through the rotation of the segments, the pressure torque is increased to reduce the demand for friction. The wheel requires the smaller friction force, so its obstacle surmounting performance is higher. Especially in the case of load, the effect of reducing the demand for friction will be further amplified. At the same time, the rotation of the segment will more closely fit the obstacle surface, making the obstacle-surmounting process more stable. Furthermore, due to structural, the actual contact conditions of obstacle surmounting will be different. In the case of $C_2$ of segment, it is better than the circular wheel in the whole process of surmounting obstacles; while in the case of $C_1$ of segment, when the obstacle reaches a certain height, it has the effect of saving mechanical work. As the height gradually rises, the effect of mechanical work saving will also become more and more obvious.

The transformable wheel [15] of this type can improve its obstacle surmounting performance by more than 10% under normal road conditions by calculating the friction ratio between the transformable wheel and the circular wheel. Different from its linkage mechanism, makes its wheel diameter become larger, the transformable wheel proposed in this study does not surmount the high obstacle with its pivoting-head part rotating freely. So theoretically this transformable wheel can improve its performance by nearly 30% with its wheel torque and drive force under the road conditions of low-height obstacles, and is especially suitable for carts and suitcases passing through low-height obstacles such as blind lanes and low stairs. It is expected that its effect will be more pronounced under load.

However, this study still has some problems. First, the transformable wheel in this study is composed of the same pivoting-head part divided into ten equal parts. In order that each segment can rotate freely without interference, in wheel state, the wheel surface is a discontinuous curve, and the rolling speed is limited to a certain extent. Second, under the specifications of this study, the obstacles below 8 mm cannot be explained by the theoretical model. As a future subject, we will continue to surmount these.

**Author Contributions:** Methodology, Y.C.; design and experiment, A.K.; writing—review, M.I.; writing—review and coaching for design, J.I.; review and coaching for data acquisition Y.S. All authors have read and agreed to the published version of the manuscript.

**Funding:** This research was funded by Comprehensive Research Institute in Tokyo Denki University, grant number Q21D-06.

**Institutional Review Board Statement:** We exclude this Statement.

**Informed Consent Statement:** Not applicable, and we exclude this Statement.

**Data Availability Statement:** Not applicable.

**Acknowledgments:** We got lots of help from students who are in Iwase Lab and Inoue Lab in Tokyo Denki University.

**Conflicts of Interest:** The authors declare no conflict of interest.

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
