# Peer review of "Design and Analysis of Transformable Wheel with Pivoting-Head Mechanism"

_2673-3161, doi:10.3390/applmech4010005_

Round 1

Reviewer 1 Report

A transformable wheel is designed to meet the requirements of traveling different road conditions. And this kind of wheel can used to pass through low-height obstacles, which can serve to facilitate the travel of middle-aged and elderly people. The paper can be accepted after considering the following recommendations.

(1)   The contribution of this study should be highlighted in the “Introduction” section. And whether there are studies with similar structures, the authors should emphasize about the advantages of the structure proposed in this paper.

(2)   There are several issues regarding the discussion of Figure 1 as follows. (1). What are the units of h in it; (2). What does it mean that the small orange and green squares are in different orientations at different h? The reviewer hopes the author can explain it in detail.

(3)   The values of a,b,r,θ0 in Figure 7 should be explained.

(4)   The authors should add some contents to their discussion, for example: within the discussion in Figure 8, the red line has an inflection point when the value of δ becomes large, and the authors should discuss this phenomenon. Also, the value of this inflection point changes when the obstacle height becomes a different value, and the authors could explain the reason for this and compare it with the results of previous literatures.

(5)   Questions about the experimental equipment: the way of measuring torque, the brand and accuracy of the measuring equipment are suggested to be explained.

(6)   For newton's representation, n should always be capitalized, such as line 452.

(7)   In the conclusion, it is mentioned that “the transformable wheel proposed in this study can improve its performance by nearly 30% under the road conditions of low-height obstacles” and a comparison graph is suggested to added.

(8)   The language of this paper needs to be embellished, for example, the last sentence of the “Conclusion” section.

(9)   Since there are too many symbols involved in this paper, it is suggested to add a symbol description table.

(10) The citation of this literature does not see a specific serial number, and it is suggested to adjust the format of the citation.

Author Response

Q(1).The contribution of this study should be highlighted in the “Introduction” section. And whether there are studies with similar structures, the authors should emphasize about the advantages of the structure proposed in this paper.

A:Benefited from this structure, the convenience of replacement can be achieved without changing the original wheel size. And it’s different from that expanding the radius of wheel to improve the obstacle surmounting performance in the past, Without changing the size, it can focus more on the obstacle surmounting performance of low-height obstacles, and improves its performance ratio, and pay more attention to the performance of load situation ._(1)

Q(2).There are several issues regarding the discussion of Figure 1 as follows. (1). What are the units of h in it; (2). What does it mean that the small orange and green squares are in different orientations at different h? The reviewer hopes the author can explain it in detail.

A:

Under the situation of the actual contact, the parts of the tire that are in contact with the obstacle may be C1 or C2. And in order to facilitate the subsequent analysis and calculation, the contact conditions are only listed in the standard state and the limit state. By integrating the contact conditions in different situations (the contact point is on C1 or C2, and whether the wheel state is the standard or limit state), when dealing with specific  obstacle surmounting problems, it can be in the form of a similar way which is look-up table with backward inference method, to infer the situations of wheel contacting. (2)And the limit status (the limit positions of C1 and C2) has an adjustment range of 36. Based on the posture of the Figure 3, the state reflected in the Table 2 is the opposite. According to the Table 2, for obstacles of the same height, the contact position of the segment is different from the posture of the transformable wheel when contacting. So we can determine the position of the contact segment unit by the height of the obstacle to facilitate subsequent analysis and calculation (the included angle in the next Section is δ). Above it, we can use the parameters, that the specific segment unit contact position and its rotation angle, to improve the range accuracy.

Q(3).The values of a,b,r,θ0 in Figure 7 should be explained.

A:According to the analysis in Section 3.2, beside the constant parameters of structural design such as radius and mass, the final mechanical parameters will be affected by the posture angle θ and δ. Therefore, according to the existing analysis and constraint conditions, we can obtain value range of the posture angle θ and δ. During the process of obstacle surmounting, the segment units aren’t fixed, and it will turn a certain angle with the shape of the obstacle. At this time, the pressure of the transformable wheel from the obstacle will not pass through the shaft center, and the pressure moment LN will be generated, as shown in Figures 3 and 4, based on Expression (16) and (30),_(3) we express it as Expression (36). The maximum value of the distance L from the obstacle to the center of the wheel is the radius r(3) of the transformable wheel, based on Expression (17) and (31),_(3) which is expressed as Expression (37). Based on Expressions (7), we need to ensure that there is an actual obstacle colliding with the transformable wheel, and express it as Expression (38). So we can obtain the following constraint condition Expressions (36 ~ 38) :

Regarding the parameters a and b in the above expression, based on the contact conditions in Figures 3 and 4, select the case where the two inequalities are all true, that is the parameter values at the intersection of the ranges of the inequalities. In Expression (38), since the height of the obstacle in contact will not change suddenly, Expression (7) can be combined with Expression (18) and (32), then θ0 can be converted into θ._(3) With the above restraint conditions, Expressions (36 ~ 38), we can obtain the value range by the coordinate whose the abscissa is θ and the ordinate is δ, as shown in the red shaded part in Figure 7. Above this, the specific range can be determined according to the obstacle height.

Q(4).The authors should add some contents to their discussion, for example: within the discussion in Figure 8, the red line has an inflection point when the value of δ becomes large, and the authors should discuss this phenomenon. Also, the value of this inflection point changes when the obstacle height becomes a different value, and the authors could explain the reason for this and compare it with the results of previous literatures.

A:The image information in Figure 14~16, which is consistent with the described situation in the section of torque arm L and wheel torque M. By observing the figures in Section 3.3.2, the red curves which is contacted with C1 all appear V-shape, and combined with the corresponding calculation formula, it is found that the contact section of C1 has the composition of Equation (43). After calculation, there is a situation that the formula under root will be less than 0, so we simplify it to the size of 1 and sinθ0+sin(γ+δ)+sinδ. In its equivalent case, it is the extreme value of the V-shaped inflection point. At the same time combining Figure 4 and Equation (11), Figure 17 can be obtained. The actual meaning of Equation (43) is the comparison of s1 and s3. When the two are the same, it is the extreme value of the inflection point. At the same time, when the height h(s1) is low, the proportion of s2 is relatively large, and the segment only needs to rotate slightly to generate the s3, that is, as shown in Figures 8, 11, and 14, the distance between the extreme points and the connection points by red and green curves is close; on the contrary, when the height h(s1) becomes larger, the proportion of s2 becomes smaller, and the rotation of the segment will become larger, so that the corresponding s3 can be generated, that is, as shown in Figures 10, 13, and 16, the distance between the extreme points and the connection points by red and green curves is far._(4)

Q(5).Questions about the experimental equipment: the way of measuring torque, the brand and accuracy of the measuring equipment are suggested to be explained.

A:

In this experiment, the transformable wheel is placed on a bench with variable height, and its rotating shaft is connected with a tension gauge, as shown in Figure 18. In the case of sufficient friction, the horizontal pull can be regarded as its Drive force. When we pull the force gauge horizontally,_(5) the transformable wheel will surmount the obstacle, then we record the data on the force gauge at this time. The force gauge used in this experiment is AIKOH Model 9800 Series as shown in Figure 19. Its measurement accuracy is ±0.2%F.S, and the smallest unit of measurement is 0.01N, and the measurement range is 0~50N._(5) We record the required the force of the wheel state and the transformable state in Table 3.

Figure 19. Force gauge AIKOH Model 9800 Series_(5)

Q(6).For newton's representation, n should always be capitalized, such as line 452.

A:

Above 8mm steps in Figure 15~16, we can find the corresponding point to 2.81N in 10mm and 3.6N_(6) in 15mm. At this time, there are two posture angles in C1 and one posture angle in C2, which meet the above conditions. Through the actual contact situation, the corresponding posture angle can be obtained.

Q(7).In the conclusion, it is mentioned that “the transformable wheel proposed in this study can improve its performance by nearly 30% under the road conditions of low-height obstacles” and a comparison graph is suggested to added.

A:

The transformable wheel [15] of this type can improve its obstacle surmounting performance by more than 10% under normal road conditions by calculating the friction ratio between the transformable wheel and the circular wheel_(7). Different from its linkage mechanism, makes its wheel diameter become larger, the transformable wheel proposed in this study doesn’t surmount the high obstacle with its pivoting-head part rotating freely. So theoretically this transformable wheel can improve its performance by nearly 30% with its wheel torque and drive force_(7) under the road conditions of low-height obstacles, and is especially suitable for carts and suitcases passing through low-height obstacles such as blind lanes and low stairs. Expected, its effect will be more pronounced under load._(8)

Q(8).The language of this paper needs to be embellished, for example, the last sentence of the “Conclusion” section.

A:

The transformable wheel in this study is developed by the characteristic that the pivoting-head parts generate the torque that the pressure doesn’t pass through the wheel axis by the wheel transforming. Through the rotation of the segments, the pressure torque is increased to reduce the demand for friction. The wheel requires the smaller friction force, so its obstacle surmounting performance is higher. Especially in the case of load, the effect of reducing the demand for friction will be further amplified. At the same time, the rotation of the segment will more closely fit the obstacle surface, making the obstacle surmounting process more stable. And due to structural, the actual contact conditions of obstacle surmounting will be different. In the case of C2 of segment, it is better than the circular wheel in the whole process of surmounting obstacles; while in the case of C1 of segment, when the obstacle reaches a certain height, it has the effect of saving labor. As the height gradually rises, its The effect of labor saving will also become more and more obvious._(8)

However, this study still has some problems._(8) First, the transformable wheel in this study is composed of the same pivoting-head part divided into ten equal parts. In order that each segment can rotate freely without interference, in wheel state, the wheel surface is a discontinuous curve, and the rolling speed is limited to a certain extent. Second,_(6) under the specifications of this study, the obstacles below 8mm can’t be explained by the theoretical model. As a future subject, we will continue to surmount these

Q(9).Since there are too many symbols involved in this paper, it is suggested to add a symbol description table.

A:Table 1. The parameter symbols used in this manuscript_(9)

Symbols

Description

Symbols

Description

r

the radius of round wheel

h

Stair height

A

rotation point of segment

B

Contact point

N

Pressure on a segment part

F

Friction on a segment part

MN

Pressure torque

MF

Friction torque

M

Wheel torque

L

Distance from the center of the rotation shaft to the stair

a

The abscissa of the center of the transformable wheel

b

The ordinate of the center of the transformable wheel

X

The abscissa of the center of the contact point

Y

The ordinate of the center of the contact point

θ0

The angle within friction and stair

γ

The rotation angle of segment center

δ

The angle between OA and X axis

Q

Drive force

Reviewer 2 Report

A figure in addition to figure 1 will be helpful in understanding the structure and working principle sections clearly.

The position of the text for the X, and Y coordinates in figures 2, 4, and 5 can be moved, as currently, it is not clearly visible

Grammatical error, line 91

Some more details about the experimental setup and procedure shown in figure 2 will be helpful for the reader to better understand the methods used.

Author Response

â… .A figure in addition to figure 1 will be helpful in understanding the structure and working principle sections clearly.

A:Figure 1. Pivot fixing and releasing mechanism(a&b: Sectional view; c: Demo )_(â… )

â…¡.The position of the text for the X, and Y coordinates in figures 2, 4, and 5 can be moved, as currently, it is not clearly visible

A:Figure 2. Mechanical analysis of round wheel (Left: force diagram; Right: actual contact)_(â…¡&â…£)

Figure 4. Mechanical analysis of C1 section in deformation state (Left: force diagram; Right: actual contact)_(â…¡&â…£)

Figure 5. Mechanical analysis of C2 section in deformation state (Left: force diagram; Right: actual contact)_(â…¡&â…£)

â…¢.Grammatical error, line 91

A:Clark et al. [9] presented a transformable wheel, Adabot, which can smoothly be converted from a round wheel, to a wheel with tire studs, to a legged-wheel. Wheel transformations are performed by extending wheel struts radially outward from the center of the wheel. Adabot has been optimized using an evolutionary algorithm such that its physical characteristics and its controller are better able to handle terrain that includes obstacles of varying sizes. Yu et al.[10] presented a novel wheeled robot that transforms from a circled configuration to a spoke-like legged configuration. In general wheeled mobile robots are able to move quickly and efficiently on flat surfaces._(â…¢) The robot’s wheel is comprised of five spokes or legs, four of which are actively driven by a motor via four slider-crank linkages. The last one is designed be passive in order to significantly decrease the actuation force of the transformer mechanism. The diameter of the legged configuration wheel is 1.576 times that of the circled configuration; Kim et al. [11] reported a new wheel-leg hybrid robot. This robot utilizes a novel transformable wheel that combines the advantages of both circular and legged wheels. With the new transformable wheel, the robot can climb over an obstacle 3.25 times as tall as its wheel radius, without compromising its driving ability at a speed of 2.4 body lengths/s with a specific resistance of 0.7 on a flat surface.

â…£.Some more details about the experimental setup and procedure shown in figure 2 will be helpful for the reader to better understand the methods used.

A:Figure 2. Mechanical analysis of round wheel (Left: force diagram; Right: actual contact)_(â…¡&â…£)

Figure 4. Mechanical analysis of C1 section in deformation state (Left: force diagram; Right: actual contact)_(â…¡&â…£)

Figure 5. Mechanical analysis of C2 section in deformation state (Left: force diagram; Right: actual contact)_(â…¡&â…£)
